

# Optimizing WRF-Hydro Calibration in the Himalayan Basin: Precipitation Influence and Parameter Sensitivity Analysis

5      Ankur Dixit[1,2,*], Sandeep Sahany[3], Flavio Lehner[2,4,5], Saroj Kanta Mishra[1]

[1]Centre for Atmospheric Sciences, Indian Institute of Technology Delhi, India

[2] Department of Earth and Atmospheric Sciences, Cornell University, Ithaca, NY, USA

[3]Climate Projections and Extremes Branch, Centre for Climate Research Singapore, Singapore

10     [4]Climate and Global Dynamics Laboratory, National Center for Atmospheric Research, Boulder, CO 80305

[5]Polar Bears International, Bozeman, MT 59772

[*]Corresponding Author – ad933@cornell.edu



## Abstract

Lack of in-situ observations and reliable climate information in large part of the Himalayas poses a significant challenges for assessing water resources vulnerability accurately. Further, the reliance on only a few coarse resolution gridded datasets with considerable uncertainty complicates this problem. However, integrated hydrometeorological modeling systems, like WRF-Hydro, have the potential to provide information in such ungauged or poorly observed regions, yet they require careful calibration. Here, we calibrate and assess the fidelity of WRF-Hydro in simulating the hydrological regime of the Beas basin. Selected WRF-Hydro model parameters are calibrated using the PEST framework, using eight simulations with two meteorological forcings from two WRF realisations. Model calibration improves the accuracy of the basin discharge simulation, however the choice of precipitation forcing is also critically important. We propose an ensemble weighting scheme to optimize an intra-annual tradeoff between streamflow under- and overestimation in different WRF-Hydro configurations. This study demonstrate the efficacy of using coupled (offline) WRF and WRF-Hydro for providing climate change impact-relevant information in data-sparse basins.

**Keywords**: WRF-Hydro Calibration; PEST; Parameter sensitivity;  Parameter optimization; Model inversion

35

## Summary

This study calibrates WRF-Hydro in a Himalayan basin, finding precipitation choice significantly influences results over parameter sets. Study highlights the importance of tailored calibration strategies and parameter sensitivity analyses for accurate streamflow predictions in Himalayan basins, crucial for effective water resource management.



## 1. Introduction

The Himalayan ranges host one of the largest ensembles of glaciers that provide freshwater supply to most of South- and Southeast-Asia. However, water shortage is a rising concern in this region due to global warming causing snow/ice cover depletion. Climate change affects the cryosphere and the mountainous water cycle, i.e., glaciermelt and snowmelt, streamflow, precipitation, and runoff seasonality (IPCC AR6 WGII). Policymakers require a climate-resilient transition framework and pathways for efficient water resource management that further requires risks estimation through assessment of hazards and vulnerabilities. Threfore, a sustainable and adaptive framework necessitates reliable information on historical and projected water resources.

The response of water security in the region to climate change is intricately woven into a complex interplay involving climate change dynamics, glacier reduction, water availability, and evolving water demand patterns. (Cogley, 2011; Gardelle et al., 2012; Scherler et al., 2011; Maurer et al., 2019; Nie et al., 2021). The water demands of the Himalayas could be met at least by the end of $21^{st}$ century with water currently stored in glaciers (W. W. Immerzeel et al., 2013; W. W. Immerzeel and Bierkens, 2012) and large changes in runoff are unlikely anytime soon (T. Bolch et al., 2012; Khanal et al., 2020). Even so, the Beas basin is reportedly under stress of increasing water demand and could face severe water shortage (Kumar et al., 2007; Moors et al., 2011).

Moreover, the risk of insufficient streamflow to serve agriculture, horticulture, tourism, and hydropower is growing under climate change (Dixit et al., 2023; Dar et al., 2014; Murtaza and Romshoo, 2015; Romshoo et al., 2015; Scott et al., 2012; Slingo et al., 2005). Around 35% of the upper Beas basin's average annual flow is contributed by snow and glacier melt during 1990-2004 (Kumar et al., 2007; Moors et al., 2011) that further reportedly increased to 52-56% during 1996-2008(Ahluwalia et al. 2015). Shean et al., (2020) found that the excess glacier melt runoff (resulting in a negative mass balance) to be in the range of ~12-53% for each basin in High Mountain Asia. In addition, Dixit et al. (2021) reported that ~90% of glaciers in this region could disappear by $2094 \pm 3.5$ years under RCP4.5, and by $2084 \pm 8$ years under RCP8.5. Therefore, enough evidence is available to believe that the vulnerability of snow and glacier-based water resources in this region cannot be ignored.

However, inconsistent/unreliable observations along with poor in-situ coverage pose a challenge to close the water budget in this region (Hewitt, 2005; Bolch et al., 2012; Hartmann and Andresky, 2013; Maussion et al., 2011; Li et al 2017). Moreover, gridded observations tend to have coarse spatial resolution (for example IMD, APHRODITE, TRMM, PERSIANN-CDR, CPC, CHIRPS, CRU and GPCP; Dixit et al. 2023). Alternatively, deploying a numerical model at convection permitting scale may fill up this gap, in particular given recent improvements in capturing localized features over complex terrain (Collier et al. 2013; Rasmussen et al. 2014). Therefore, coupling (online or offline) of climate impact models with high-resolution numerical models can provide useful insights about present and projected





future changes. But, climate impact models (such as hydrological models or agriculture models) need calibration or tuning of its parameters before deployment.

Accuracy of streamflow simulations from such models is affected by the model calibration strategy, including optimization methods, target parameters, and calibration/validation period (Bittelli et al., 2009; Givati et al., 2012; Pennelly et al., 2014; Ragettli and Pellicciotti, 2012; Silver et al., 2017; Yucel et al., 2015). Shortlisting of target parameters is important and often depends on the basin's characteristics. Overall, calibration of a hydrological model is region specific and a necessary exercise before its deployment. However, calibration inherits complexity from the model structure and definition of the involved processes. Estimation of parameters involved in physical processes require extensive expertise that may not always be readily available in the existing literature (Arsenault et al., 2014; Beven and Freer, 2001; Duan et al., 1992, 2006; Gan et al., 2014; Kavetski et al., 2003; Sorooshian and Gupta, 1983).

Literature highlights usage of several methods to estimate these parameters and quantify their uncertainty, such as markov chain monte carlo (MCMC), monte carlo sampling, stratified sampling, Gauss-Marquardt-Levenberg algorithm, and bayesian networks (Matott et al., 2009). Several studies also preferred to perform manual adjustment to these parameters to get an optimum set. However, manual adjustment has limited search capability in solution space that may prevent obtaining a global minima or the best parameter set, whereas automatic methods are capable of exhaustive searching in parameter space, making these methods very popular recently (Arsenault et al., 2014; Moradkhani and Sorooshian, 2008; Tolson and Shoemaker, 2007). For example, heuristic or semiheuristic approaches can find the global minimum by itself through iterative searches (Arsenault et al., 2014).

Distribution prior and posterior to calibration is an important feed for automatic methods to optimize the parameter set. Various methods use either one or both of them to adjust parameters (Jin et al., 2010). Some of the most frequently used algorithms are generalized likelihood uncertainty estimation (GLUE) method (Beven and Binley, 1992), Metropolis–Hastings (MH) algorithm, and a Markov Chain Monte Carlo (MCMC) method (Bates and Campbell, 2001; Beven and Freer, 2001; Freer et al., 1996; Kuczera and Parent, 1998). In this study, we demonstrate the capability of PEST (Parameter ESTimation), a model-independent parameter estimation tool, to calibrate WRF-Hydro with an exhaustive parameter search utilizing two modes, estimation mode and regularisation mode.

The aim of this study is to calibrate WRF-Hydro with an automatic approach as an alternative of manual adjustment method. The objectives are (1) to identify the sensitivity of various parameters for a Himalayan mountain headwater basin, (2) explore the feasibility of calibrating a wide parameter set (considering parameters beyond just soil parameters i.e. vegetation parameters or snow parameters), especially for regions like the Beas basin with shallow soil depth and dense vegetation cover, and (3) evaluate and find the best method for the calibration. This study also extends previous work with PEST and WRF-Hydro that used either a limited number of parameters or idealized representation of some of the processes such as no



infiltration of retention of water (Li et al. 2017; Wang et al. 2019; Senatore et al. 2015). In addition, it serves as a case study for calibrating WRF-Hydro for both operational use and climate change studies over complex terrain.

## 1.1 WRF-Hydro modelling overview

WRF-Hydro is a spatially distributed, physically based hydrologic model that can be coupled with an atmospheric model in online or offline mode. Details on the model's basic structure can be found in Supplementary section S2 or the model's technical description (https://ral.ucar.edu/projects/wrf_hydro/documentation).

Several studies calibrated WRF-Hydro over different parts of the world using manual and automatic methods. Wang et al. (2019) calibrated WRF-Hydro using PEST over the midwestern United States (US) for Manning's coefficient (MannN), saturated hydraulic conductivity (REFDK), runoff/infiltration rate (REFKDT), and channel slope. Sofokleous et al. (2023) conducted a grid based calibration of WRF-Hydro for 31 small mountain watersheds in Cyprus using three parameters infiltration, hydraulic conductivity and percolation. Liu et al. (2021) tested the parameter sensitivity of REFKDT, retention depth (RETDEPRT), surface roughness (OVROUGHRT), and MannN over semi-humid and semi-arid areas of northern China. Li et al. (2017) evaluated the performance of WRF-Hydro over Beas basin and found it reasonably well over this region. They calibrated REFKDT, soil evaporation exponent, soil layers, saturated soil hydraulic conductivity, reference soil moisture for transpiration, RETDEPRT, OVROUGHRT, and MannN. Past calibration efforts of the model suggest a wide range of parameters in consideration, most of them are soil or channel parameters. In this study, we consciously broaden the calibration parameter set to include vegetation parameters as this region has a shallow soil column but significant vegetation cover. One of the goals is to determine if model performance can be further improved by incorporating vegetation parameters in calibration process.

## 1.2 PEST (Parameter ESTimation) framework

PEST is a parameter optimization framework and newly introduced in surface hydrological models, but otherwise widely used in groundwater modelling. It performs model inversion to optimize its parameters using advanced mathematical concepts and transformations.

PEST calculates the parameter sensitivity through parameter optimization iterations, governed by various thresholds. Each optimization iteration starts with an initial sensitivity determined by the Jacobian matrix based on the parameters' initial values. Thereafter, model runs are performed with updated parameter sets, generated using control parameters. PEST's parallelization capability can perform multiple runs (in isolation) at a time to reduce overall time cost. Two parallel framework are available: parallel-PEST and BeoPEST. They are similar



to perform their job as both of them parallelize the PEST, but BeoPEST has a better runtime
manager and better parallelizing capability. More details about PEST can be found in
Supplementary Section S3-S5 and Doherty (2016).

## 2 Data and Methods

## 2.1 Study Area

Beas basin is bounded by the outer, middle, and greater Himalayan ranges, and situated in
Himachal Pradesh state of India (Figure 1). The microeconomics of the region is majorly driven
by agriculture/horticulture and tourism. The Beas river is also an important tributary of the
Indus river system and hence plays a significant role to fulfil the water demand downstream.

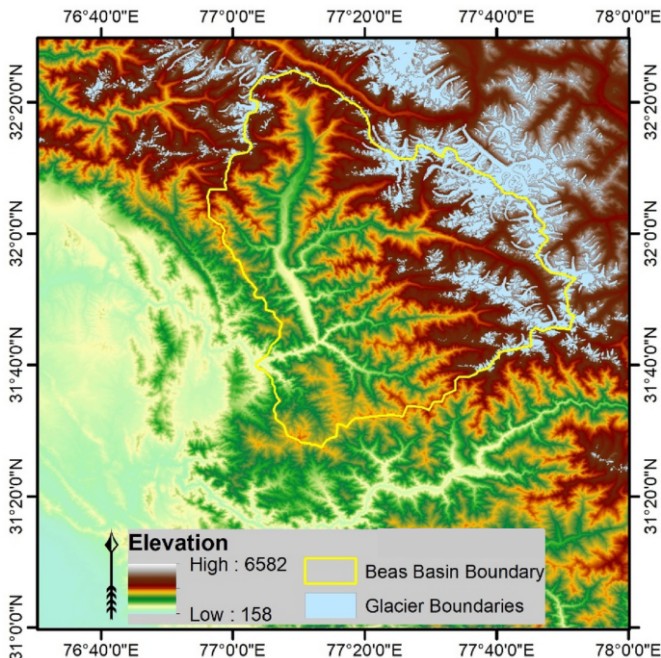

Figure 1: Beas basin with basin boundary, underlaid by a Digital Elevation Model (SRTM
90 m). The highest and lowest elevation within the basin boundary is approximately 6545 masl
and 826 masl, respectively. The sky blue region represents glaciers. Glaciers in study region
cover approximately 12.6% of the area.

Sharp changes in topographical conditions and diverse physiography makes this region very
diverse in spatio-temporal conditions. The topography varies from 826 to 6545 masl (Hegdahl
et al. 2016), with more than 20% area above 4800 masl and the highest peak at more than





6500 masl. The upper reaches have deciduous and alpine vegetation, while the lower reaches
have a substantial amount of area under agricultural and horticultural practices. Soild
precipitation occurs mostly during the winters because of the Western Disturbances (WDs),
contributing significantly to the riverstream during summer. The Indian Summer Monsoon
(ISM) causes mostly rainfall and has the highest variability to the annual cycle of precipitation
and streamflow (55% vs. ~ 7%; Kumar et al. 2007).

## 2.2 Observational Dataset

Sparse coverage of in-situ observation makes model verification a challenging task in this
region. However, intercomparison of available gridded datasets and past literature provides
some basis to select the most reliable among them (Dixit et al., 2023). APHRODITE has more
spatial heterogeneity and better spatial variability over higher elevations compared to other
gridded datasets (TRMM: Tropical Rainfall Measuring Mission, APHRODITE: Asian
Precipitation—Highly-Resolved Observational Data Integration Towards Evaluation,
CHIRPS: Climate Hazards Group InfraRed Precipitation with Station, GPCP: Global
Precipitation Climatology Project, PERSIANN-CDR: Precipitation Estimation from Remotely
Sensed Information using Artificial Neural Networks- Climate Data Record, IMD: Indian
Meteorological Department, and CPC: Climate Prediction Center; Dixit et al. 2023; Figure 2).

APHRODITE is also preferred by other studies as an observational dataset over the Himlayan
regions (Andermann et al. 2011), having a reasonable temporal variation of precipitation
(Ghimire et al. 2018). We also rely on APHRODITE as a reference dataset; refer to Dixit et al.
2023 for more details, as they also compared these observation datasets with a few in-situ
observations in the Beas basin and concluded that APHRODITE is the best among the selected
datasets.

Streamflow observations at the outlet (Thalout) provide a reference output for model
calibration. As per availability, we obtain 3 year of observational data (2003-2005) to use as
calibration and validation references.



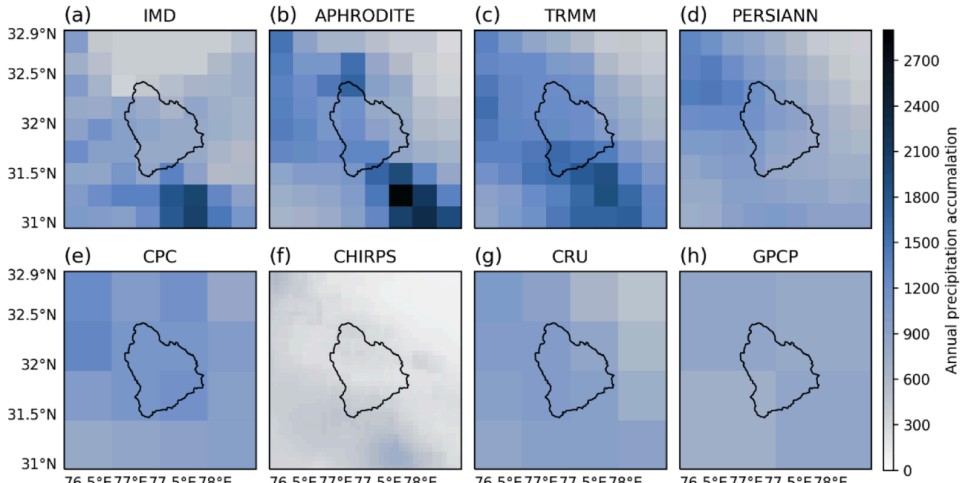

Figure 2- Total annual precipitation of 2003 for (a) IMD, (b) APHRODITE, (c) TRMM, (d) PERSIANN-CDR, (e) CPC, (f) CHIRPS, (g) CRU and (h) GPCP.

## 2.3 WRF setup and simulations

WRF configuration (including parameterizations and forcing conditions) is adopted from Dixit et al. (2023) to generate high-resolution meteorological forcing to feed WRF-Hydro. Only

exception to the configuration is LULC updation, as MODIS LULC underestimate glaciers significantly (Figure 3-b). It happens because of debris deposition over glacier surfaces causing a change in its reflectance profile shifting towards barren rock. High resolution satellite image (Figure 3-a) shows most of the glacier area being misclassified as debris. However, updated LULC shows glaciated area closer to the observation (RGI V6 glacier boundaries; Figure 3-

c,d)

ERA-Interim reanalysis dataset from European Centre for Medium-Range Weather Forecasts (ECMWF) and customized by the National Center for Atmospheric Research (NCAR) (ds627.0|DOI: 10.5065/D6CR5RD9) provided initial and boundary conditions to the model. Dixit et al. (2023) performed sensitivity tests of microphysics and cumulus schemes over this

region and found that no scheme performed well in producing the precipitation throughout the annual cycle; however, MP8KF (Microphysics: MP8; Cumulus: KF) and WSM6BMJ (Microphysics: WSM6; Cumulus: BMJ) performed reasonably well for December-January (DJF) and June-September (JJAS), respectively. Therefore, two sets of simulations, each with MP8KF and WSM6BMJ are performed.



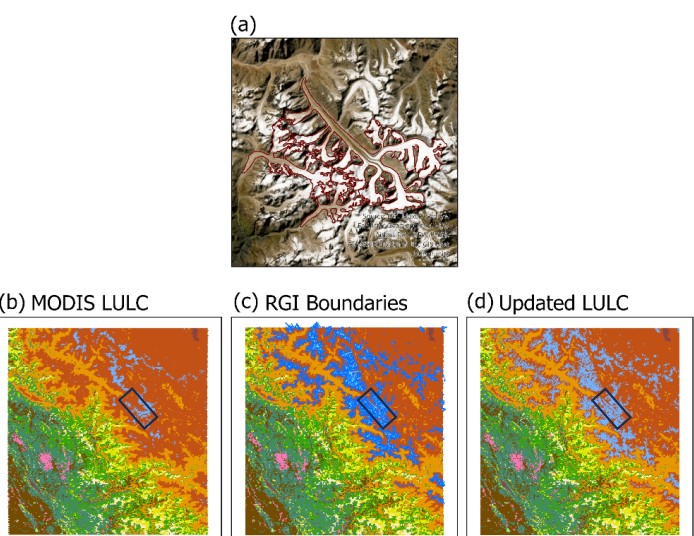

Figure 3– The glacier image from the MODIS LULC dataset used in the WRF-Hydro configuration. The blue shades in b-d represents the glacial area. (a) A zoomed-in high-resolution satellite image (LANDSAT 8) of the Bada Shigri glacier, with the RGI V6 glacier boundary overlay. The debris on the glacier makes it challenging to differentiate between glacier and barren rock. (b) The default MODIS LULC dataset. (c) The MODIS LULC dataset overlaid with RGI V6 glacier borders from the region. (d) the current MODIS LULC dataset. A black rectangular rectangle in panels b-d depicts the approximate position of a glacier in panel a.

## 2.4 WRF-Hydro setup

WRF-Hydro is offline-coupled with WRF and forced with hourly WRF outputs for precipitation, near-surface air temperature, humidity, surface pressure, incoming shortwave radiation, incoming longwave radiation, and wind speed. There is no feedback from the hydrological model to the atmospheric model. NCAR's GIS tool is used to outline basin boundaries, stream networks, and routing link files utilizing high resolution DEM and outlet location (https://github.com/NCAR/wrf_hydro_arcgis_preprocessor/archive/v5.1.1.zip).

WRF-Hydro configuration requires the setup of two components- land surface components and routing components. Land surface options are mostly inherited from land surface model i.e. Noah-MP in this case. Noah-MP is a one-dimensional multi-physics parameterized model that can simulate various terrestrial fluxes related to energy, water, snow, and soil. The selected schemes for physical processes are listed in Table 1.



Table 1 – The selected options in the Noah-MP multiphysics parameterization for the physical
processes.

| Physical Process | Option |
|---|---|
| Dynamic vegetation option | LAI/SAI from lookup table<br>Max vegfrac from climatology |
| Canopy stomatal resistance | Ball-Berry |
| Soil moisture factor for stomatal resistance | Noah |
| Surface layer drag coefficient | M-O |
| Frozen soil permeability | Niu and Yang (2006) |
| Super cooled liquid water | Niu and Yang (2006) |
| Radiative transfer | Two-stream applied to vegetated fraction |
| Ground snow surface albedo | CLASS |
| Precipitation partitioning | Jordan (1991) |
| Lower boundary condition for soil temperature | Bottom temperature from climatology |
| Snow/soil temperature time scheme | Semi-implicit with FSNO |
| Surface resistance to evap/sublimation | Snow/non-snow split |
| Glacier treatment | Slab |

Aggregation factor in WRF-Hydro provides a flexibiltity to represent the soil state parameters
at higher resolution by disaggregating the parameters as per AGGFACTRT value. We chose
this factor as 10 to get the disaggregated parameters at grid size of 300 m. Additionally, the
model has switches to activate/deactivate different routing schemes such as overland flow,
channel flow, and subsurface flow.

Here, the D8 algorithm is used to activate the routing for both overland and subsurface flflow,
allowing surplus water from one grid to flow in the direction of its neighboring eight grids'
steepest slope. Channel routing is solved using diffusive wave gridded. It simplifies St. Venant
equations for shallow water waves and is a one-dimensional diffusive wave with variable time-
steps. It formulates channel flow by integrating the diffusive waves using a first-order Newton-
Raphson equation; however, in low gradient cases—which do not exist in our study area—it
may cause certain instabilities. The baseflow estimate is performed using the exponent bucket
model. Six seconds of timestep integration is used for routing schemes. To a depth of one
meter, the soil column is separated into four vertical layers.




## 2.5 WRF-Hydro calibration

WRF-Hydro simulations obtained their meteorological forcing from WRF model output and terrestrial boundary conditions from geogrid file produced by WRF Preprocessing System (WPS). For calibration experiments, the model is initialized on 01 October 2001 and ran until 31 December 2003, having year 2002 as a spinup period. Station-observed streamflow (Thalout) is used as a reference dataset to calibrate (2003), and validate (2004-2005) the model output. Calibration experiments follow the WRF output feeding to WRF-Hydro repeatedly under the PEST framework (Supplementary Figure S1).

Parameter constraints are important during the calibration process to reduce computational cost/time and preserve the physical significance of the model. Moreso, selection of parameters is a priority to avoid computational time/resources overhead. Therefore, One-At-a-Time (OAT) sensitivity is performed for 108 parameters from HYDRO.TBL, MPTABLE.TBL, GENPARM.TBL, and CHANPARM.TBL having each parameter perturbed to its default value in a model run. These parameters are selected based on dominant processes from different components of WRF-Hydro and available literautures. For example, Arsenault et al. (2018) found that five soil and six (or more) vegetation parameters control the sensitivity of sensible heat, latent heat, and soil moisture in Noah-MP. Vegetation parameters also play a key role to define a coupling of water and energy exchange between surface and atmosphere.

OAT performs a single model run with a single parameter perturbation to find a parameter's sensitivity based on a change in the hydrograph. Thereafter, the 42 most sensitive parameters are selected to undergo composite sensitivity analysis and model inversion (Table 2). Parameters' bounds are defined from the literature and previous studies.

Table 2 – The sensitive parameters chosen after one-at-a-time sensitivity analysis. The violet color denotes channel parameters, yellow soil parameters, green biophysical parameters, and cyan snow parameters.

| Parameter | Initial | Min | Max | | Parameter | Initial | Min | Max |
|-----------|---------|-----|-----|---|-----------|---------|-----|-----|
| chslp1 | 4.134077 | 1.5 | 4.5 | | omega2 | 0.4 | 0.25 | 0.75 |
| chslp2 | 1.057639 | 0.5 | 2 | | betads | 0.5 | 0.25 | 0.75 |
| chslp3 | 0.151969 | 0.1 | 0.65 | | betais | 0.5 | 0.25 | 0.75 |
| chslp4 | 0.3 | 0.05 | 0.3 | | z0sno | 0.002 | 0.0005 | 0.02 |
| mn1 | 0.95 | 0.1 | 0.95 | | rsurf | 50 | 20 | 150 |
| mn2 | 0.387234 | 0.1 | 0.8 | | saiw1 | 0.4 | 0.15 | 0.9 |
| mn3 | 0.283335 | 0.03 | 0.65 | | saim1 | 0.35 | 0.15 | 0.9 |
| mn4 | 0.100935 | 0.03 | 0.45 | | sais1 | 0.6 | 0.35 | 0.9 |



| | | | | | | | |
|---|---|---|---|---|---|---|---|
| refdk | 2.48E-06 | 1.00E-07 | 4.25E-05 | saiw5 | 0.4 | 0.2 | 1 |
| refkdt | 3.124642 | 1.5 | 7.5 | saim5 | 0.5 | 0.2 | 1.5 |
| zbot | 5.69325 | 2 | 8 | sais5 | 1 | 0.5 | 2.5 |
| zvt1 | 3.05 | 1 | 8 | saiw7 | 0.2 | 0.1 | 0.5 |
| zvt5 | 0.8 | 0.35 | 2.5 | saim7 | 0.2 | 0.1 | 0.5 |
| zvt7 | 0.06 | 0.025 | 2 | sais7 | 0.7 | 0.3 | 1.5 |
| hvt1 | 20 | 10 | 60 | laiw1 | 4 | 1 | 7 |
| hvt5 | 16 | 8 | 30 | lais1 | 4 | 1 | 7 |
| hvt7 | 1.1 | 0.5 | 7 | laiw5 | 2 | 1.5 | 4.5 |
| mfsno | 2.5 | 0.2 | 20 | lais5 | 3.5 | 2.5 | 7.5 |
| albic1 | 0.8 | 0.6 | 0.95 | laiw7 | 0.2 | 0.05 | 0.5 |
| albic2 | 0.55 | 0.4 | 0.75 | lais7 | 2.3 | 1.5 | 5.5 |
| omega1 | 0.8 | 0.55 | 0.95 | lvcoef | 0.5 | 0.25 | 0.75 |

However, model inversion with 42 parameters create a 42-dimensional solution space that may
be computationally expensive. Thereby, SVD is utilized to perform parameter variance
selection to reduce the dimensionality that reduce identifiability of Parameters (PI) because of
their limited variance in truncated solution space. The PI of various parameters is assessed
using the four singular values 25, 30, 36, and 42 (Figure 4). Notably, solution space lost the

variance and hence the identifiability of known to be sensitive parameters in truncated solution
space with 25 and 30 singular values (Figure 4; channel parameters). However, most of these
parameters are identifiable with 36 singular values but this is not more helpful to reduce the
cost significantly. Therefore, regularisation is implemented while using all 42 parameters.
Regularization can eliminate non-useful parameters to the null space in a particular iteration

during the inversion process. However, they can return back to the solution space if they
become sensitive in the following iterations.

(a) MP8KF Experiments          (b) WSM6BMJ Experiments

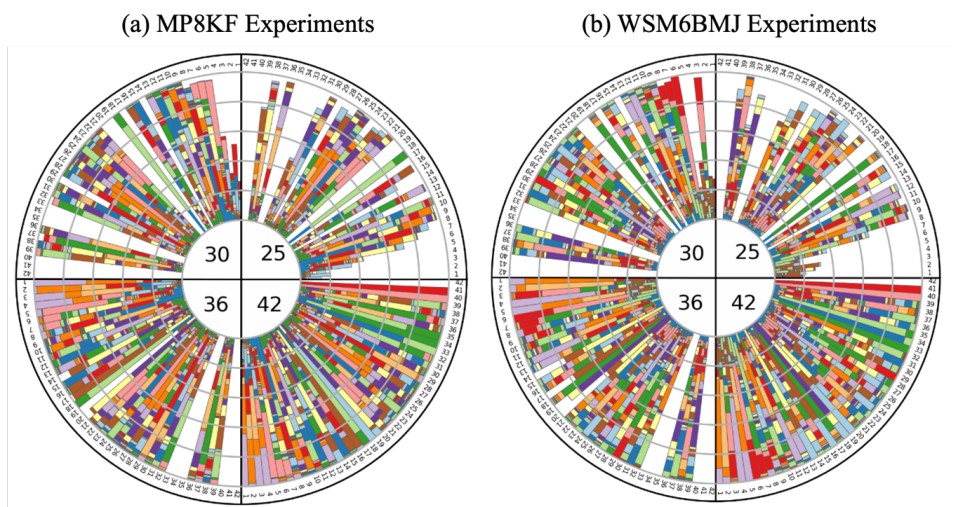

Figure 4– The parameter identifiability (PI) plot for all 42 parameters, with 25, 30, 36, and 42 singular values. (a) Displays PI using the MP8KF forcing; (b) Displays PI using the WSM6BMJ forcing.

## 3. Results

### 3.1 WRF evaluation (Precipitation)

WRF simulated precipitation underestimate observation over the Himalayan foothills of the Beas basin when evaluated using six experiments: all combination of three microphysics schemes: MP3, MP8, and WSM6; and two cumulus schemes: KF, abd BMJ, (Dixit et al. (2023)). The most reasonable combination was reported to be BMJ cumulus scheme along with WSM6 or MP8 microphysics. They concluded that MP8KF is better than the rest of the experiments in simulating precipitation for annual cycle (ANN) and DJF. However, for JJAS, WSM6BMJ simulates better variance, pattern correlation, temporal correlation and skill score than MP8KF, making WSM6BMJ a more favorable option to simulate JJAS climate. More details about these simulations, results, and discussion is available in Dixit et al. (2023).

### 3.2 WRF-Hydro evaluation (streamflow)

### 3.2.1 MP8KF* Experiments

These experiments consistently underestimate streamflow at its outlet especially during the ISM (JJAS). However, they follow observation reasonably well during the winter (Figure 5). Uncalibrated WRF-Hydro produces the least accurate streamflow (Table 4). Calibration improved JJAS streamflow in all experiments for both the calibration as well as validation





period (especially for year 2005), although improvement is not sufficient with respect to the observed discharge. Performance metrics (NSE=Nash–Sutcliffe Efficiency; KGE=Kling-Gupta Efficiency; d=index of agreement) quantify the visual improvement in the hydrograph: NSE increases from 0.1 to 0.3 (with regularisation experiments) during the calibration period and from 0.16 to ~0.4 for the validation period (Table 4). KGE and *d* also improved for both

calibration and validation period (Table 4).

Regulariation experiments show better performance in comparison to the classical estimation; however regularised SVD performs the best among these experiments for calibration.

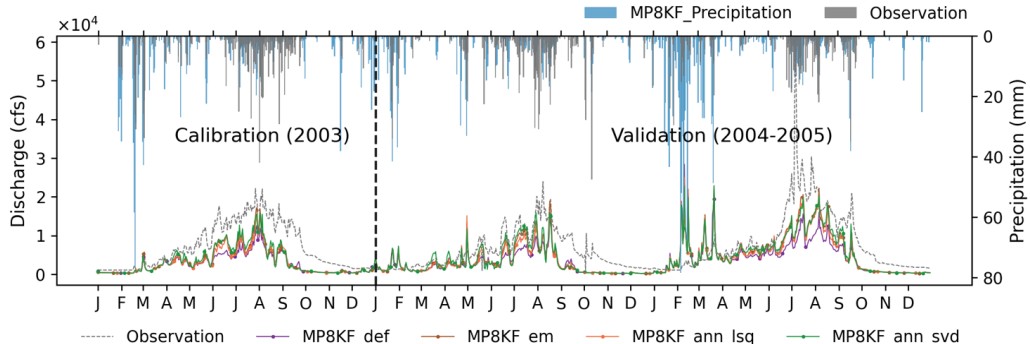

Figure 5 – Comparison of PEST-calibrated/validated discharge (with MP8KF forcing) in estimate and
regularization mode, with default model discharge and observed discharge at the basin terminal. The top x-axis and right y-axis depict precipitation from APHRODITE and the driving WRF simulation (MP8KF forcing).

Table 4 – The accuracy metrics for calibrated/validated WRF-Hydro discharge (using MP8KF forcing) with observation. The metrics are produced separately for calibration (2003) and validation (2004-2005). d: index of agreement; KGE: Kling-Gupta Efficiency; NSE: Nash-Sutcliffe Efficiency Coefficient; RMSE: root mean squared error.

| Experiment / Accuracy | CALIBRATION (2003) | | | | VALIDATION (2004-2005) | | | |
|---|---|---|---|---|---|---|---|---|
| | d | KGE (2009) | NSE | RMSE | d | KGE (2009) | NSE | RMSE |
| MP8KF_def | 0.67 | 0.15 | 0.01 | 5714.74 | 0.65 | 0.24 | 0.16 | 6126.81 |
| MP8KF_em | 0.75 | 0.30 | 0.21 | 5109.82 | 0.76 | 0.40 | 0.35 | 5396.66 |
| MP8KF_reg_lsq | 0.72 | 0.25 | 0.15 | 5293.15 | 0.79 | 0.46 | 0.40 | 5146.30 |
| MP8KF_reg_svd | 0.78 | 0.35 | 0.30 | 4783.72 | 0.78 | 0.45 | 0.39 | 5200.29 |




### 3.2.2 WSM6BMJ* Experiments

WSM6BMJ* experiments show good representation of the summer precipitation totals, but
tends to overestimate precipitation during other parts of the year (Fig. 6, top x-axis); thus the
two WRF simulations, MP8KF and WSM6BMJ, have largely opposing strengths and
weaknesses over the course of a calendar year. Consequently, WRF-Hydro calibration under
WSM6BMJ* experiments shows substantial improvement in the hydrograph for summer
streamflow in comparison to MP8KF* experiments (compare Fig. 5 and 6). However, these
experiments produce spurious peaks during winter and spring corresponding to heavy storms
in precipitation (Figure 6). Despite that, NSE shows reasonable accuracy (0.45; higher than
each of the MP8KF experiments) for the default setup. Calibration inproves NSE to 0.53 along
with improvements in KGE and d (Table 5).

Similar to the previous section, SVD produces better accuracy for calibration. Performance
metrics show that these experiments perform almost similar for both calibration and validation
period.

Among the calibration techniques, we find that SVD with regularised inversion generally
produces the highest accuracy (Table 5), though a calibration using estimation mode is slightly
better for validation period (Table 5).


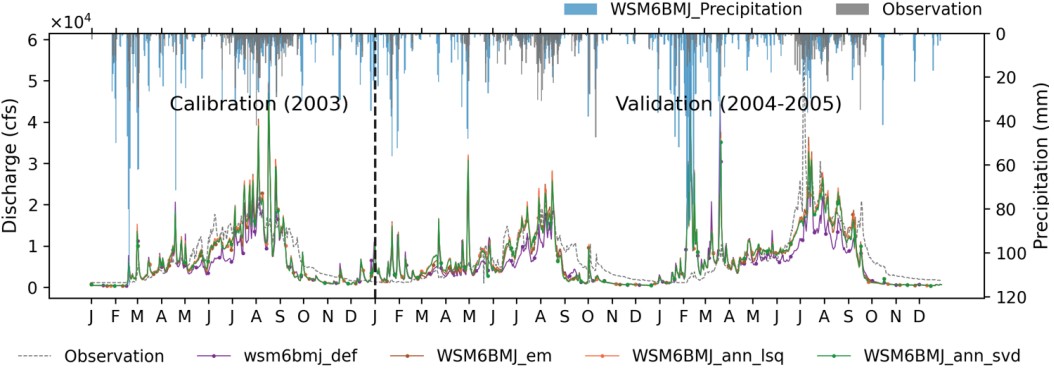

Figure 6– Comparison of PEST-calibrated/validated discharge (with WSM6BMJ forcing) in estimate
and regularization mode with default model discharge and observed discharge at the basin outlet. The
top x-axis and right y-axis show precipitation from APHRODITE and WRF simulation (WSM6BMJ
forcing).



Table 5 – The accuracy metrics for calibrated/validated WRF-Hydro discharge (with WSM6BMJ forcing) with observation. The metrics are produced separately for calibration (2003) and validation (2004-2005). D: Index of agreement; KGE: Kling-Gupta Efficiency; NSE: Nash-Sutcliffe Efficiency Coefficient; RMSE: Root Mean Squared Error.

| Experiment / Accuracy | CALIBRATION (2003) | | | | VALIDATION (2004-2005) | | | |
|---|---|---|---|---|---|---|---|---|
| | d | KGE (2009) | NSE | RMSE | d | KGE (2009) | NSE | RMSE |
| WSM6BMJ_def | 0.86 | 0.69 | 0.45 | 4258.94 | 0.75 | 0.54 | 0.26 | 5723.76 |
| WSM6BMJ_em | 0.89 | 0.69 | 0.48 | 4126.01 | 0.84 | 0.71 | 0.40 | 5172.67 |
| WSM6BMJ_reg_lsq | 0.89 | 0.69 | 0.50 | 4066.78 | 0.83 | 0.69 | 0.37 | 5287.68 |
| WSM6BMJ_reg_svd | 0.90 | 0.72 | 0.53 | 3930.53 | 0.82 | 0.69 | 0.38 | 5262.51 |


### 3.2.3 WSM6BMJ (JJAS) Experiments

The previous two sections show that the performance of the calibrated model depends strongly on the season, most likely because of the precipitation differences in the respective experiments. Therefore, one might expect a seasonally specific calibration to further improve model performance during that season. To test this, we conduct additional experiments with the same configuration as in Section 3.2.2, but only for JJAS.

In line with previous sections, the SVD method demonstrates superior performance in comparison to the LSQ method in terms of accuracy of calibrated discharge. LSQ exhibited inadequate accuracy and poorly reconstructed discharge (Figure 7, Table 6). Despite model performance metrics not indicating an ideal match of the SVD method with observations, a substantial improvement is achieved from its default configuration. However, it is worth noticing that both WSM6BMJ_ann_svd and WSM6BMJ_jjas_svd exhibit almost similar performance for calibration. Validation findings suggest that WSM6BMJ_ann_svd possesses a slight advantage, an outcome that was unexpected but can be attributed to various factors. For instance, the validation runs do not commence in June every year, instead, they encompass full year runs with WSM6BMJ_ann_svd/WSM6BMJ_jjas_svd calibration parameters, so that the model is not getting an accurate initial condition at the start of the summer in WSM6BMJ_jjas_svd. Another possible explaination is that the model was allowed to run across different season, not just summer, using season-specific calibration parameter, potentially leading to inconsistent fluxes and inaccurate feedback.



Overall, both SVD experiments yield similar results in reproducing JJAS discharge, with no significant advantage of one method over the another. Notably, calibration for the JJAS period is relatively quicker to achieve compared to calibrating for an entire year.


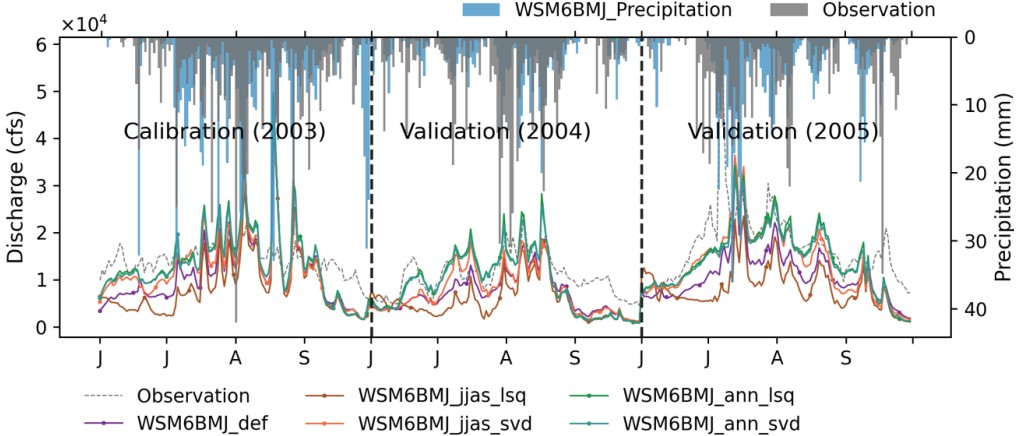

Figure 7– Comparison of PEST-calibrated/validated discharge (with WSM6BMJ forcing for JJAS only) using estimation and regularisation mode with default model discharge and observed discharge at the basin outlet. Top x-axis and right y-axis show precipitation for APHRODITE and WRF simulation
(WSM6_BMJ forcing)

Table 6 – The accuracy metrics of calibrated/validated WRF-Hydro discharge during JJAS only (with WSM6BMJ forcing) compared to observed discharge. The metrics are computed separately for the calibration (2003) and validation periods (2004-2005). d: index of agreement; KGE: Kling-Gupta
Efficiency; NSE: Nash-Sutcliffe Efficiency Coefficient; RMSE: root mean squared error.

| Experiment / Accuracy | CALIBRATION (2003) | | | | | VALIDATION (2004-2005) | | | |
|---|---|---|---|---|---|---|---|---|---|
| | d | KGE (2009) | NSE | RMSE | | d | KGE (2009) | NSE | RMSE |
| WSM6BMJ_def | 0.58 | -0.41 | -3.78 | 6468.45 | | 0.71 | 0.48 | -0.02 | 6962.52 |
| WSM6BMJ_jjas_lsq | 0.47 | -0.50 | -6.33 | 8010.38 | | 0.55 | 0.21 | -0.46 | 9349.84 |
| WSM6BMJ_jjas_svd | 0.65 | -0.31 | -2.55 | 5579.96 | | 0.80 | 0.64 | 0.17 | 6079.46 |
| WSM6BMJ_ann_lsq | 0.62 | -0.71 | -3.6 | 6341.48 | | 0.80 | 0.66 | 0.24 | 5894.49 |
| WSM6BMJ_ann_svd | 0.63 | -0.62 | -3.25 | 6099.20 | | 0.80 | 0.65 | 0.24 | 5929.34 |




### 3.2.4 Ensemble of estimations

WRF-Hydro inherits the performance intricacies from WRF's precipitation input. Any extreme
change in precipitation can lead to a similar behavior in discharge. MP8KF* underestimates
summer precipitation/discharge but simulates the rest of the year relatively well. WSM6BMJ*
produces summer precipitation/discharge better than MP8KF* but simulates the rest of the year
relatively poorly. Therefore, seasonal dependent sensitivity of WRF/WRF-Hydro's
precipitation/discharge simulation motivates the use of ensembles to obtain reliable
information.

Hence, we compute an ensemble discharge using an experiment from MP8KF* and
WSM6BMJ* experiments. SVD experiments produced higher accuracy (section 3.2.1-3.2.3)
for calibration/validation, therefore, we use  MP8KF_reg_svd, WSM6BMJ_reg_svd, and
WSM6BMJ_jjas_svd experiments to compute an ensemble mean as well as a weighted
ensemble mean. We computer a total of four ensembles, two ensembles mean and two weighted
ensembles, using Eqs. 1-4.

**Ensemble Mean:**

$$MP8WSM\_reg\_svd\_ens\,(t) \;=\; \left(\frac{MP8KF\_reg\_svd\,(t)\,+\;WSM6BMJ\_reg\_svd\,(t)}{2}\right)$$

460                                                                                 Equation 1

$$MP8WSM\_jjas\_svd\_ens\,(t) \;=\; \left(\frac{MP8KF\_reg\_svd\,(t)\,+\;WSM6BMJ\_jjas\_svd\,(t)}{2}\right)$$

Equation 2

**Weighted Ensemble:**

$$MP8WSM\_reg\_svd\_ensW\,(t) = \begin{cases} MP8KF\_reg\_svd\,(t) & if \;\; t \notin JJAS \\ WSM6BMJ\_reg\_svd\,(t) & if \;\; t \in JJAS \end{cases}$$

Equation 3



$$MP8WSM\_jjas\_svd\_ensW\ (t)\ = \begin{cases} MP8KF\_reg\_svd\ (t) & if\ \ t \notin JJAS \\ WSM6BMJ\_jjas\_svd\ (t) & if\ \ t\ \in JJASS \end{cases}$$

470                                                                                          Equation 4

All four ensembles produce discharge with reasonable accuracy, having NSE greater than or equal to 0.5 (Table 7). The ensemble mean has both, a better mean discharge as well as greatly reduced spurious peaks outside of summer, creating a more realistic simulation overall (Figure
8). For calibration, NSE increases to 0.56 (for weighted ensemble mean) and 0.64 (for ensemble mean) which is higher than any individual experiment in section 3.2.1-3.2.2 (Table 5-7). However, validation suggests the weighted ensemble has a better NSE (0.60) than ensemble mean (0.47). Nevertheless, both calibration and validation accuracy improve following the ensemble approach (Figure 8, Table 7). The validation period (2004-2005) has
more extreme peaks in precipitation/discharge due to winter and spring storms, affecting the accuracy of the ensemble mean, however swapping the JJAS discharge from both experiments in weighted ensemble lead to better accuracy.

Since the accuracy of the ensemble mean decreases during the validation period, the unweighted ensemble approach might not be a reliable methods. However, any of the weighted
ensemble performs better than individual experiments.

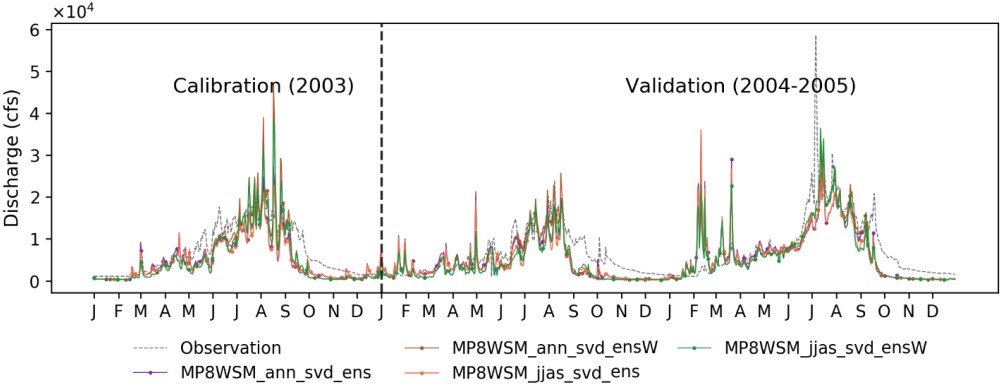

Figure 8– Comparison of the default model discharge and the observed discharge at the basin outlet with an ensemble of PEST-calibrated/validated discharge (using MP8KF and WSM6BMJ forcing)
employing estimate and regularization mode.




Table 7 – The accuracy metrics of a calibrated/validated WRF-Hydro discharge ensemble (with MP8KF
and WSM6BMJ forcing) compared to observed discharge. The metrics are produced separately for
calibration (2003) and validation (2004-2005). d: index of agreement; KGE: Kling-Gupta Efficiency;
NSE: Nash-Sutcliffe Efficiency Coefficient; RMSE: root mean squared error.

| Experiment / Accuracy | CALIBRATION (2003) | | | | VALIDATION (2004-2005) | | | |
|---|---|---|---|---|---|---|---|---|
| | d | KGE (2009) | NSE | RMSE | d | KGE (2009) | NSE | RMSE |
| MP8WSM_ann_svd_ens | 0.90 | 0.67 | 0.64 | 3430.56 | 0.83 | 0.62 | 0.47 | 4871.02 |
| MP8WSM_ann_svd_ensW | 0.90 | 0.63 | 0.50 | 4039.98 | 0.89 | 0.72 | 0.60 | 4246.53 |
| MP8WSM_jjas_svd_ens | 0.88 | 0.60 | 0.61 | 3600.79 | 0.82 | 0.59 | 0.47 | 4875.58 |
| MP8WSM_jjas_svd_ensW | 0.90 | 0.67 | 0.56 | 3780.81 | 0.88 | 0.67 | 0.58 | 4316.89 |

WRF-Hydro calibration with PEST framework in regularisation mode performs better than in
estimation mode. The objective function in the regularisation mode is complemented by an
additional cost function that prevents overfitting of the model to obtain global minima, unlike
the estimation mode that calculates the sum of squared differences between simulation and
observation. It uses Tikhonov Regularisation with SVD and LSQR solver to divide the
parameter space into the null space and solution space. The null space contains the parameters
that are inestimable, while the solution space contains the estimable parameters. The separation
of null and solution spaces makes the problem well-posed and numerically stable. To generate
more set of parameters, only estimable parameters are perturbed, however, inestimable
parameters remains fixed to their values. Applying both solutions (SVD and LSQR), we find
that SVD outperformed LSQR to provide an optimal parameter set through regularised
inversion. However, LSQR has the advantage of computational speed. Therefore, LSQR could
be a better choice for the highly parameterized inversion problem as SVD may become very
slow.



### 3.3 Parameter Sensitivity

Parameter sensitivity analysis helps to identify important parameters to undergo for calibration and perturbation. Composite sensitivity analysis assess the change in parameters' sensitivity when all parameters are perturbed at the same time. It is calculated through the Jacobian derivate or parameters' initial value. As the calibration process progresses, the values of parameter change and hence their sensitivity too. PEST does calibration process through iterations, known as optimization iterations, the value and sensitivity associated with iteration having minimum objective function known as optimized value and optimized sensitivity.

Variations in value and sensitivity over iterations shows PEST capability to search the parameter space to find an optimum value (Figure 9-10). These figures are shown for regularised inversion experiments for MP8KF and WSM6BMJ forcing using SVD and LSQ; and are not shown for other calibration experiments as they seem to show similar information.

A few biophysical parameters (saim7, sais7, zvt7, hvt7, laiw7, lais7, albic1, albic2, and lvcoef) did not change over the iterations (Figure 9). The corresponding sensitivity of these parameters also suggest no substantial role in inversion process (Figure 10). Besides, some parameters (chslp3, chslp4, sais5s) reached to their bounds and got freezed after some variations. These contrasting behaviour of parameters shows the relative importance of these parameters with respect to others, and could help to optimize the calibration process through further subselection of parameters.

Once a parameter is frozen to its bound, it no longer participates in the inversion until next optimization iteration. Channel slope parameters shows limited/no variations, except for the second-order channel slope. Numbering (1,5 and 7) in the lai, sai, hvt, and zvt parameters refer to the specific LULC class. 1 stands for Evergreen Needleleaf Forest, 5 stands for Mixed Forest, and 7 stands for open Shrublands. Most of the biophysical parameters did not show variations for open Shrublands because of the limited landscape belonging to this class.

If a parameter is not sensitive, it is not considered as an adjustable parameter, and hence no more set of values are generated. Therefore, parameters that showed almost no variations (saim7, sais7, zvt7, hvt7, laiw7, lais7, albic1, albic2, and lvcoef) are found to be insensitive. Furthermore, the channel slope parameters are sensitive (composite sensitivity) but reach to their bounds and does not participate in inversion. Because model calibration is a constrained inversion problem that means a parameter could be sensitive enough but can take a value in a certain range to not loose its physical relevance. The constrained parameter calibration approach also helps to reduce model equifinality sets, however, they can not be minimized by just constraining parameters and require more advanced multi-objective functions. Moreover, frozen parameters may have different sensitivity over the iterations due to the changes in jacobian derivative that depends on the initial sensitivity computed using parameters' initial values. A change in initial value can lead to a change in parameter's sensitivity.



Manning coefficients are one of the most sensitive parameters (Figure 10) in addition to runoff/infiltration rate (refkdt) and saturated hydraulic conductivity (refdk). A few biophysical parameters (lai, sai, hvt, zvt) associated with Evergreen Needleleaf Forest and Mixed Forest
are also sensitive. Snow parameters are sensitive except for the snow albedo parameters (albic1, albic2). Among snow parameters, mfsno is the most sensitive snow parameter.

Though many parameters have different optimum values over the iterations, most of them are closer in their sensitivity under the SVD and LSQR regularised inversion (Figure 10). This signifies the method's robustness to provide optimal sensitivity (and hence inversion solution)
to find the global minima. However, this also signifies the phenomena of equifinality (Beven and Freer, 2001; Savenije, 2001; Wang et al., 2019) where it is possible to get the many parameter sets associated with global minima. The impact of equifinality can be reduced using multi-observation objective function. For example, alongside the observed discharge, observed soil moisture or biophysical properties also constraint calibration process to limit number of
ways for a model to reach global minima.





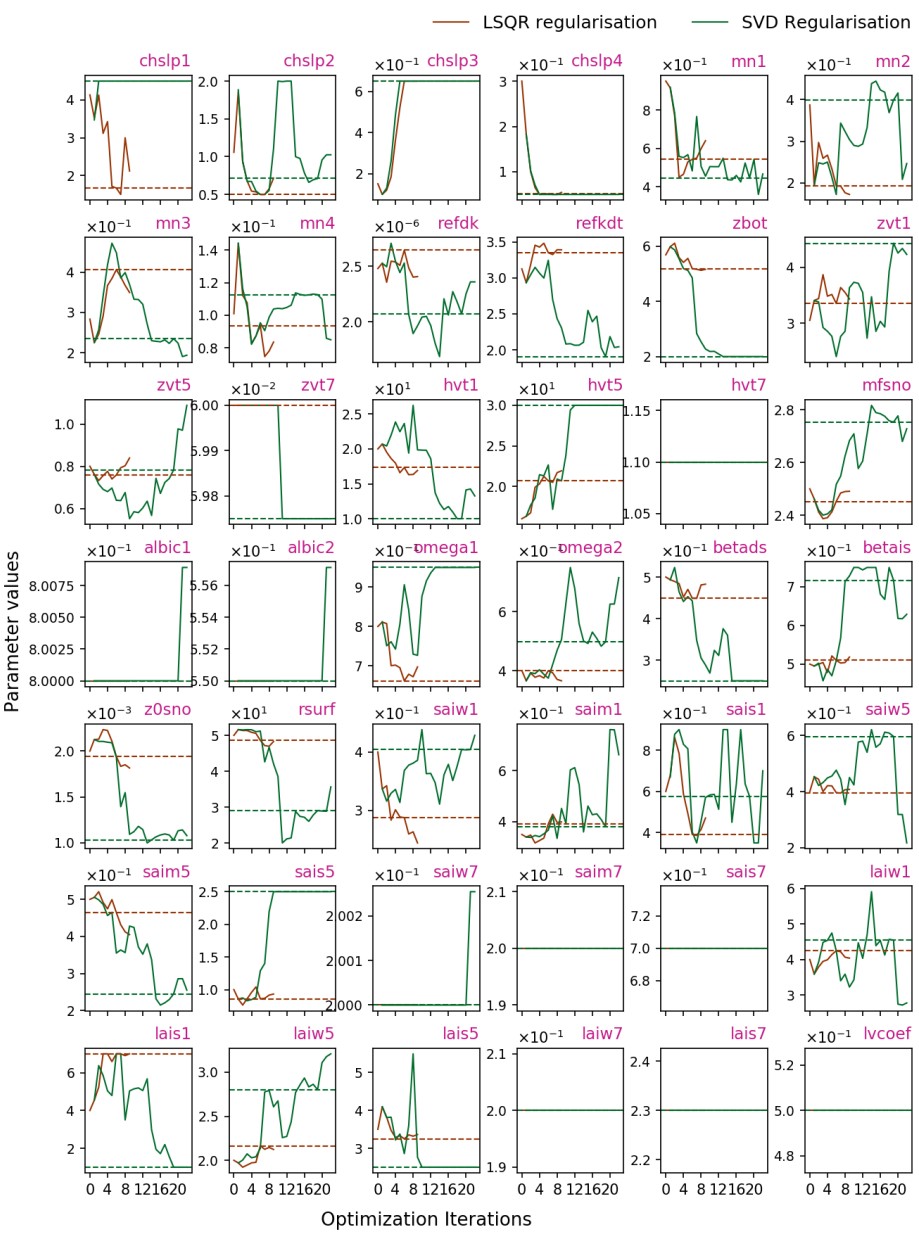

Figure 9– Variations in the parameter values of all adjustable parameters utilized in the inversion process under MP8KF forcing. The parameter fluctuations are displayed over the optimization iterations of the calibration process in regularisation mode, using SVD and LSQR inverse solutions. The solid line depicts parameter fluctuations, whereas the dashed line represents the parameter value in the most optimal iteration. The color green indicates SVD regularisation, while yellow denotes LSQR regularisation.

Figure 10– Variations in the sensitivity of all adjustable parameters utilized during the inversion process using MP8KF forcing. The composite parameter sensitivity is displayed over the optimization iterations of the calibration process in regularisation mode, using the SVD and LSQR inverse solutions. The solid line represents the parameter sensitivity, whereas the dotted line represents the parameter sensitivity in the most optimized iteration. The color green indicates SVD regularisation, while yellow denotes LSQR regularisation.



## 4. Summary and Discussion

Here we attempt to calibrate WRF-Hydro over a Himalayan basin with an automated
obtimization scheme. We find that calibration results are largely influenced by the choice of
precipitation sourced from WRF. Among the calibration experiments, it is evident that the
choice of precipitation input have more substantial influence on the outcome than the selection
of parameter sets. Consistent with Dixit et al. (2023), the MP8KF* experiments are
underestimating the discharge during summer and the WSM6BMJ* experiments are found to
have extreme spurious peaks. Most of these discharge characteristics are inherited from the
precipitation. Nonetheless, parameter tuning is also important to match the observed peaks and
contributes to improve model performance. Our findings are consistent with Li et al. (2017)
who performed hydrological simulations over the same region with WRF-Hydro fed with
WRF-derived meteorological forcing similar to our MP8KF setup. They found that discharge
is underestimated. They added additional runoff from a glacier model while keeping infiltration
and retention depth to zero to compensate the offset. This clearly helped to attain desirable
discharge but may not be helpful for other hydrological fluxes such as soil moisture or
subsurface runoff.

Instead, we find that improved precipitation can improve the calibration performance and helps
to achieve reliable streamflow. Moreover, we followed an ensemble weighting approach in this
study to overcome the underestimation of summer precipitation. Alternatively, bias correction
could also help in the regions where observations are available with sufficient frequency in
time and space, which is not the case in our region. Our study region has significant snow/ice
cover that necessitates using sub-daily forcing to reliably compute snowmelt runoff and,
consequently, streamflow. Further limitations and reasons for not performing precipitation bias
correction are discussed in the Supplementary Section S6.

Instead, the contrast between MP8KF*/WSM6BMJ* experiments and their respective
underestimation/overestimation of the summer/winter-spring precipitation helps to obtain an
accurate discharge through forming an ensemble mean that balances the counteracting effects.
WRF-Hydro calibration with PEST framework in regularisation mode performs better than its
estimation mode. However, LSQR has the advantage of computational speed. Therefore, LSQR
could be a better choice for the highly parameterized inversion problem.

Many parameters failed to help converging the model and are considered as not useful
parameters. Most of these parameters are biophysical parameters associated with open
shrubland. However parameters associated with Evergreen Needleleaf Forest (ENF) and Mixed
Forest (MF) shows substantial role in calibration. Additionally, deep soil temperature depth
(zbot), momentum roughness length (zvt1, zvt5), height of canopy (hvt1, hvt5), and snow
related parameters are found to be sensitive to produce discharge. Notably, ice albedo



parameters (albic1, albic2) are not sensitive which are important to estimate radiation fluxes and eventually melt through energy exchange with downward sensible heat flux. We suspect the semsitivity of these parameters are compromised by other parameters, especially mfsno, however further investigation is required to come out precisely with the explaination. mfsno is a snowmelt parameter and is the most sensitive snow parameter that is responsible for the snow cover fraction of a grid. The z0sno (snow surface roughness length) and rsurf (snow surface resistance) are also found sensitive to control melt discharge. The soil parameters refdk and refkdt are also sensitive to the volume of discharge through runoff/infiltration partitioning. The channel parameters are among the most influential parameters for discharge and holds highest sensitivity. Channel slope parameters are sensitive but froze to their bounds quickly after a few iterations that shows its higher sensitivity to influence discharge. Slope parameter can influence discharge volume and peak substantially. Since, JJAS discharge is underestimated in uncalibrated version of the model, PEST does try produce higher peaks through increasing slope parameters and eventually reaching its bound. Therefore, it is advise to be cautious while defining bounds for these parameters.

WRF-Hydro provides flexibility to define a scaling factor for overland flow roughness parameter (OVROUGHRT) and retention depth (RETDEPRT). It can be used to provide spatial heterogeneity of these parameters. These parameters help to increase/decrease the water availability over the surface and hence sometimes used as to calibrate the discharge (Kerandi et al., 2018; Li et al., 2017; Wang et al., 2019). Wang et al., (2019) used RETDEPRT as 0.001, which means the soil would have lesser retention depth, hence lesser ponded water. The lesser ponded water provides more water to runoff to join the channel, thereby produce more surface and channel runoff. They found that the model underestimated discharge, so infiltration was kept to zero to provide more surface water to the runoff and routing scheme. The zero infiltration would provide more water to runoff from the surface and join channel to produce more discharge. However, these scaling parameters are not changed in this study. Instead, the associated LULC type parameters are selected to go through the inversion process and find the optimal value.

## 5. Conclusions

By conducting a full sensitivity and calibration demonstration, this study shows the applicability of the calibrated version of WRF-Hydro coupled with a reliable version of WRF. It can be a useful tool to provide robust climate information for consumption of policymakers, especially to overcome poor spatio-temporal coverage of observation in this region.



However, any model calibration exercise is highly dependent on the regional physiographical and topographical conditions, so it should be an independent exercise for every region and every model before its deployment. Therefore, values of the parameters we found optimal may not be directly used for other parts of the world. However, the method to calibrate the model can be applied over any region. Moreover, a few of the most sensitive biophysical, soil, and

snow parameters identified can be used as initial targets when performing calibration in similar regions, saving time and computational cost. This study also recommends to use the regularised inversion with SVD solver for parameter optimization, consistent with Wang et al. (2019). Wang et al. (2019) did a limited time calibration of few days but found that SVD regularisation outperforms other methods over the midwestern United States. We recommed to follow this

approach for hydrological model calibration over any part of the world as PEST is model independent and can be coupled with any model with slight configurational changes.

**Competing interests**

The contact author has declared that none of the authors has any competing interests.

**Acknowledgement**

We would like to thank the MoES, Govt. of India, for providing partial financial assistance as a PhD fellowship under the IITD-MoES MoU on "Capacity building in the field of earth and atmospheric sciences" at CAS, IIT Delhi. The DST Centre of Excellence in Climate Modeling at IIT Delhi (Project RP03350) is thankfully acknowledged for logistical support. The authors thank IIT Delhi HPC facility for computational resources.


**Code Availability**

- WRF-Hydro code is available at https://github.com/NCAR/wrf_hydro_nwm_public
- WRF code is available at https://github.com/wrf-model/WRF

**Data Availability**

Softwares used in this study are open source and are free to download/use i.e. WRF
(https://github.com/wrf-model/WRF/releases), WRF-Hydro
(https://ral.ucar.edu/projects/wrf_hydro/model-code), and PEST
(https://pesthomepage.org/programs). Python is used in production of figures and other

analysis of results. Meteorological datasets are also available in open domain i.e.
Observational datasets-
    IMD (https://www.imdpune.gov.in/cmpg/Griddata/Rainfall_25_NetCDF.html),
    APHRODITE (https://www.chikyu.ac.jp/precip/english/downloads.html)
    TRMM (https://disc.gsfc.nasa.gov/datasets/TRMM_3B42_Daily_7/summary)

PERSIANN-CDR (https://chrsdata.eng.uci.edu/)
    CPC (https://psl.noaa.gov/data/gridded/data.cpc.globalprecip.html)
    CHIRPS (https://data.chc.ucsb.edu/products/CHIRPS-2.0/)



CRU ([https://crudata.uea.ac.uk/cru/data/hrg/](https://crudata.uea.ac.uk/cru/data/hrg/))
GPCP ([https://downloads.psl.noaa.gov/Datasets/gpcp/](https://downloads.psl.noaa.gov/Datasets/gpcp/))
Reanalysis datasets –
ERA-Interim ([https://rda.ucar.edu/datasets/ds627.0/dataaccess/](https://rda.ucar.edu/datasets/ds627.0/dataaccess/) )
The river discharge dataset is not publicly available but can be purchased from Bhakra Beas
Management Board (BBMB)

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
