# Peer review of "Optimizing WRF-Hydro Calibration in the Himalayan Basin: Precipitation Influence and Parameter Sensitivity Analysis"

_EGUsphere, 2024_

## Author Comment (AC1)

This paper presents the application of the PEST algorithm to calibrate the WRF-Hydro hydrologic modeling system in the Beas basin in the Himalayan region. While the paper's goals could be within the scope of the journal, I found the paper hard to understand in many parts. Therefore, I recommend rejection at this stage.

Response: We appreciate the reviewer's time and feedback on our manuscript. Acknowledging the concern regarding readability, we are committed to make necessary changes in revised manuscript to enhance the clarity and overall readability of the manuscript.

The objectives are not properly explained, and little details are provided about the hydrologic and WRF atmospheric model, the datasets, and the calibration approach. The origination is also quite confusing. For example, what is the link between Sections 1.2 and 2.5?

Response: We acknowledge the reviewer for this feedback. We did not provide much details on the model itself as it is a well known model and quite popular in the community. To keep the paper concise, we kept the model description brief, however, the necessary information about model's setup and configuration is provided in section 2.3-2.4.

Regarding datasets, we have section 2.2 to discuss the observations we used and their reliability. Section 2.3 contains the information about datasets used to force WRF model with initial and boundary conditions. Table 1 contains all the adopted necessary Noah-MP parameterization options, and Table 2 contains perturbed parameters along with their range for calibration process.

Since Section 1 is a general introduction, section 1.2 provides only a brief introduction of the PEST framework used for calibration. However, Section 2 provides the actual Data and Methods, including Section 2.5 (titled WRF-Hydro calibration), which provides details on the calibration approach/methodology of WRF-Hydro. We also provided more information about PEST in a supplementary text and cited the associated technical document.

However, we will make an effort to make these sections more readable and understandable in the revised manuscript.

In many circumstances, the authors refer to other papers (largely, Dixit et al., 2023) to justify the assumptions made and the data used in this manuscript, while they should have still properly explained the implications of these assumptions.

Response: We acknowledge this concern of the reviewer. We cited Dixit et al. 2023 mostly to refer to the selection of our parameterization and observational dataset choices, with briefly indicating the highlights. For example, we used a particular set of microphysics and cumulus schemes owing to their performance reported in Dixit et al. 2023a. Since we used the same WRF simulation output as in Dixit et al. 2023a to calibrate the hydrological model, these references are relevant and help to maintain the brevity of the paper here and keep its focus on the calibration. The reason behind

following this paper is the weighted ensemble approach over time that provides better results than any setup that uses a single parameterization scheme for microphysics and cumulus physics.

There are two papers in the references, as below, that were erroneously both cited as Dixit et al., 2023 in the main text. We will correct this by adding subscripts along with the citation.

*Dixit, A., Sahany, S., Mishra, S. K., & Mesquita, M. D. (2023a). Seasonal dependent suitability of physical parameterizations to simulate precipitation over the Himalayan headwater. Scientific Reports, 13(1), 4756.*

*Dixit, A., Sahany, S., & Mishra, S. K. (2023b). Modeling the climate change impact on hydroclimate fluxes over the Beas basin using a high-resolution glacier-atmosphere-hydrology coupled setup. Journal of Hydrology, 627, 130219.*

Acknowledging the concern of the reviewer, we will make an effort to discuss the implications of adopting prior studies' assumptions, while keeping the focus of this paper on the calibration process.

**More generally, in my opinion, testing the effectiveness of a calibration method (here, PEST) should be done in regions where there is a wealth of data. What is learned there could then, hopefully, be translated to other areas. The authors explicitly say that the precipitation data are inaccurate in their study region. So, what is the value of this effort? This should have carefully been addressed.**

Response: We understand the reviewer's concern regarding the data availability in this region and the tradeoffs it implies. However, there is a data-paucity concern in almost the entire Himalayan region, yet this region provides water supply to billions of people, necessitating efforts to predict and project possible changes despite data limitations.

Importantly, several studies using PEST in combination with WRF-Hydro or other models (predominantly groundwater models) exist, some in regions with higher data density and quality. The PEST documentation has over 40 citations as of 2024. For example, Wang et al. (2019) used a similar approach (WRF-Hydro calibration using PEST for a flood event) over the U.S. Midwest. Their study has a 3-day calibration and 12-day validation period, using sub-daily data. They show that this period and ratio of calibration to validation time is enough to explore the associated parameter sensitivity. They also reported substantial improvement in model accuracy when using PEST. There are several other studies that calibrated WRF-Hydro for specific event classes, such as Silver et al. (2017) for flood events in arid regions. Fersch et al. (2020) calibrated WRF-Hydro for 3.5 months (15 April–31 July 2015) as a compromise between the number of model runs (about 2000), which are required during hypercube sampling and PEST optimization. They reported reasonable results in terms of Nash–Sutcliffe and volumetric efficiencies. These studies used a smaller number of parameters and shorter duration for calibration than what we do here. In short, testing of PEST has been reported extensively in the literature already, warranting our application and testing in new regions such as the Himalayas. Fundamentally, careful calibration studies in data sparse regions have always been important to pave the way forward for the hydrologic modeling community. Using PEST with WRF-Hydro helps to explore more parameter search space than some traditional and more manual methods. Our objective in this study is to provide a

methodological insight on WRF-Hydro calibration with limited data to study water availability for climate time-scale. We will add paragraphs in both introduction and discussion sections to highlight the existing literature around PEST applications, as described above, and better emphasize the added value of our study in that space.

A few more responses on the choice of data for our study:

We selected a year with average-precipitation for the calibration process, and years with both average and below-average precipitation for the validation. Our findings indicate that a model calibrated with average precipitation also produces satisfactory accuracy with below-average precipitation, attesting to the fidelity of the calibrated model across a range of annual precipitation percentiles. As often with hydrologic calibration, more data would be better, but this anomalous precipitation year at least constitutes a useful out-of-sample test. Furthermore, we calibrated and tested a range of methods and precipitation realizations, totalling ~7000 model years, which itself is a significant computational effort, which would grow exponentially with extended time periods.

Regarding precipitation data, we emphasized the observed precipitation data uncertainty not its inaccuracy, a prevalent issue across the entire Himalayan region (*Bolch et al. 2012; Hartman et al. 2013; Hewitt 2005; Li et al. 2017* ). Available gridded observations over the region are inconsistent, though APHRODITE has been reported to be the most reliable (*Andermann et al. 2011; Yatagai et al. 2012; Ménégoz et al. 2013; Palazzi et al. 2013; Li et al. 2017; Ghimire et al. 2018; Dimri et al. 2013; Mathison et al. 2013; Mishra et al. 2019*). Thus, we presented the uncertainty among observation datasets and argued why it is more reasonable to use APHRODITE as a reference dataset (see Dixit et al. 2023). Also, these observations are not used to force the hydrology model rather used to select the most reasonable set of choices for parameterization schemes.

*Andermann, C., Bonnet, S. & Gloaguen, R. (2011). Evaluation of precipitation data sets along the Himalayan front. Geochem. Geophys. Geosyst. 12(7), Q07023*

*Bolch, T. et al. (2012). The state and fate of Himalayan glaciers. Science 336(6079), 310–314.*

*Dimri, A. P. et al. (2013). Application of regional climate models to the Indian winter monsoon over the western Himalayas. Sci. Total Environ. 468, S36–S47*

*Dixit, A., Sahany, S., Mishra, S. K., & Mesquita, M. D. (2023). Seasonal dependent suitability of physical parameterizations to simulate precipitation over the Himalayan headwater. Scientific Reports, 13(1), 4756.*

*Fersch, B., Senatore, A., Adler, B., Arnault, J., Mauder, M., Schneider, K., ... & Kunstmann, H. (2020). High-resolution fully coupled atmospheric–hydrological modeling: a cross-compartment regional water and energy cycle evaluation. Hydrology and Earth System Sciences, 24(5), 2457-2481.*

*Ghimire, S., Choudhary, A. & Dimri, A. P. (2018). Assessment of the performance of CORDEX-South Asia experiments for monsoonal precipitation over the Himalayan region during present climate: Part I. Clim. Dyn. 50, 2311–2334.*

*Hartmann, H. & Andresky, L. (2013). Flooding in the Indus River basin - A spatiotemporal analysis of precipitation records. Glob. Planet. Change 107, 25–35.*

*Hewitt, K. (2005). The Karakoram anomaly? Glacier expansion and the 'elevation effect,' Karakoram Himalaya. Mt. Res. Dev.*

*Li, L., Gochis, D. J., Sobolowski, S. & Mesquita, M. D. S. (2017). Evaluating the present annual water budget of a Himalayan headwater river basin using a high-resolution atmosphere-hydrology model. J. Geophys. Res. Atmos. 122(9), 4786–4807.*

*Mathison, C. et al. (2013). Regional projections of North Indian climate for adaptation studies. Sci. Total Environ. 468, S4–S17*

*Ménégoz, M., Gallée, H. & Jacobi, H. W. (2013) Precipitation and snow cover in the Himalaya: From reanalysis to regional climate simulations. Hydrol. Earth Syst. Sci. 17(10), 3921–3936.*

*Mishra, S. K. et al. (2019). Past and future climate change over the Himalaya-Tibetan Highland: Inferences from APHRODITE and NEX-GDDP data. Clim. Change 156, 315–322.*

*Palazzi, E., Von Hardenberg, J. & Provenzale, A. (2013). Precipitation in the Hindu-kush Karakoram Himalaya: Observations and future scenarios. J. Geophys. Res. Atmos. 118(1), 85–100.*

*Silver, M., Karnieli, A., Ginat, H., Meiri, E., & Fredj, E. (2017). An innovative method for determining hydrological calibration parameters for the WRF-Hydro model in arid regions. Environmental modelling & software, 91, 47-69.*

*Wang, J., Wang, C., Rao, V., Orr, A., Yan, E., & Kotamarthi, R. (2019). A parallel workflow implementation for PEST version 13.6 in high-performance computing for WRF-Hydro version 5.0: A case study over the midwestern United States. Geoscientific Model Development, 12(8), 3523-3539.*

*Yatagai, A. et al. (2012). APHRODITE: Constructing a long-term daily gridded precipitation dataset for Asia based on a dense network of rain gauges. Bull. Am. Meteorol. Soc. 93(9), 1401–1415.*

**Finally, the level of English should be significantly improved.**

Response: We acknowledge the reviewer's suggestion, and will put efforts to improve the language to enhance readability and clarity in conveying our research and findings effectively.

---

## Author Comment (AC2)

The paper entitled "Optimizing WRF-Hydro Calibration in the Himalayan Basin: Precipitation Influence and Parameter Sensitivity Analysis" introduces the topic of deterioration of future fresh water resources in the Himalayan mountain range. The authors evaluate the use of atmospheric model WRF and hydrological model WRF-Hydro as numerical tools to assess water resources in the observations-sparse region of the Beas basin. Specifically, the authors: (1) evaluate different techniques of the PEST software to calibrate the hydrological model WRF-Hydro , (2) perform a sensitivity analysis of the parameters of the WRF-Hydro model, (3) evaluate the precipitation impact of two WRF model parameterization schemes on streamflow simulated by the WRF-Hydro model, and (4) suggest an ensemble averaging method for the WRF schemes to improve the WRF-Hydro simulations of streamflow. This paper has the potential to be a substantial contribution to the scientific community of water resources studies because it suggests methods and provides findings for improved hydrological simulations in data-sparce regions. It has a good structure regarding the presentation of methods and results, which can however be improved through clarifications in the methodology and modifications and further explanations in the presentation of the results, discussion and conclusions.

Response: We appreciate the reviewer's time and feedback on our manuscript, and really thanks for the detailed and insightful comments. We are grateful for your recognition of its potential contribution in the water resources community, specifically improving hydrological simulations in data-sparse regions. Acknowledging the suggestions regarding readability, we are committed to make necessary changes in revised manuscript to enhance the clarity and overall readability of the manuscript.

**Major comments**

**Comments on the topic of the sensitivity of parameters and calibration algorithms.**

**#1 The authors present the parameter sensitivity objective first in the presentation of the objectives of the study (L114-115) and the sensitivity analysis results last in the order of presenting the results of the paper. However, a reasonable sequence of an atmospheric-hydrological modelling study usually begins, both in Methods and Results sections, with setting up the models and the presentation of the sensitivity analyses, before the actual simulations. The authors could make this adjustment, which would make the follow-up presentation of atmospheric – hydrologic simulations easier, considering the use of different calibration algorithms in the sensitivity and the calibration.**

Response: Thanks for the suggestion. We presented non-linear sensitivity of parameters after simulations, highlighting their contributions to model convergence during calibration experiments. However, we agree with the reviewer that presenting sensitivity analysis before simulations could enhance clarity for readers. Therefore, we will revise the manuscript accordingly.

**# 2 The parameter sensitivity and calibration methods comprise an important part of the paper, yet they are not well introduced in Section 2.5. Some sentences that explain the two**

methods are mixed in the results (e.g. L505-513 and L521-527). In particular, the SVD method is first mentioned in L306 (Methods) and LSQ method in L409 (Results). Are sensitivity and calibration performed simultaneously? Are SVD and LSQ part of the regularisation mode? How regularization is defined compared to the estimation mode? What are the meanings of the LSQ and SVD abbreviations? The supplement contains some information, but the main paper should contain information as well because the calibration algorithms are evaluated throughout the paper. Please make all these concepts clear in the method section so that they are clear when the reader comes to the results.

Response: We thank the reviewer for the suggestion. Our approach to parameter sensitivity involves two steps. First, we performed linear sensitivity analysis (One-At-a-Time) before calibration simulations. Then, we carried out nonlinear sensitivity analysis (All-At-a-Time or we call it composite sensitivity) during calibration experiments.

Since the dimension of solution space is high, we attempted to reduce its dimensionality using a parameter identifiability test. This test reveals the relative contribution of each parameter in truncated solution space. However, we found that reducing the dimensions below 30 (from 42) causes some parameters to lose their variance, even though these parameters are known very well to be sensitive and influential to the hydrograph. Moreover, because the composite sensitivity is nonlinear, it can change with each optimization iteration, i.e. parameters may exhibit different sensitivity with a completely new set of values for parameters.

Therefore, we applied regularisation to automate this task, i.e. when a parameter is not sensitive in a particular optimization iteration, it is moved to the null space and excluded from the optimization process in the particular iteration. To implement regularisation, a penalty function and/or dimensionality reduction technique can be used alongside objective function. SVD and LSQR are such methods to implement dimensionality reduction during optimization iteration while Tikhonov regularization is applied to assist cost function alongside measurement objective function.

The details about regularisation and its advantage over estimation mode are provided in the supplementary text. We agree with the reviewer that, being the central concepts of the paper, these details should be explained in the main text. Therefore, we will incorporate above explanations in the revised version of the manuscript.

**# 3 Are SVD and LSQ used in previous studies? If yes, how they compare to the current study? This could be added in the discussion section.**

Response: Thanks for asking. We are not aware of other studies using SVD and LSQ regularisation and its comparison with the estimation mode, except for Wang et al. (2019), who utilized SVD regularisation to achieve numerical stability of the solution but did not detail its performance relative to other methods. However, we will revisit the literature to ensure that we have not missed any relevant studies. If we find any, we will explain and discuss them in the discussion session of the revised version of the manuscript.

**# 4 What is the conclusion on the comparison of the estimation and regularization mode in PEST? Is the performance improvement worth to use the additional computation resources required by the regularization mode? Was there a significant improvement with the regularization mode? This can be added in discussion or conclusions.**

Response: We appreciate the question. We provided a performance comparison of regularised inversion with traditional estimation mode in section 3.2. However, we may have missed objectively comparing these methods and summarizing in the discussion section to convey the message more precisely. We clearly find regularisation technique to be superior to estimation mode and provide significant improvements. We will discuss these aspects in the discussion section of the revised manuscript.

**Comments on the comparison of the WRF and WRF-Hydro configurations**

**# 5 In terms of total amounts, can the authors present the precipitation and streamflow totals in the same units (e.g. mm) for all experiments? It could also be more informative to compute and present the percent bias of modeled streamflow relative to the observed streamflow. This information can be added in the existing tables.**

Response: Thanks for the suggestion. We will add the total amounts of precipitation and streamflow, along with bias in percentage, in the revised manuscript.

**# 6 The authors state, "The performance of the calibrated model depends strongly on the season" (L403). They also assume that "one might expect a seasonally specific calibration to further improve model performance during that season" (L405-406). However, no seasonally dependent evaluation metrics are presented to justify this statement in sections 3.2.1 – 3.2.2. Could the authors present some metrics to justify the additional experiments for the summer months?**

Response: Thanks for raising this question. We conducted season-specific calibration with the expectation that it would further reduce the objective function and improve the model prediction, due to the fact that it would have lesser variance and fewer data points to fit into the solution space.

However, our findings do not confirm this hypothesis. Statistics pertaining to the June-September (JJAS) period, as discussed in sections 3.2.1-3.2.2, are provided in Table 6. For clarity, these JJAS statistics are provided for the simulation with best accuracy discussed in sections 3.2.1-3.2.2. We would emphasize the metrics and our hypothesis for conducting these experiments in the motivation section of revised manuscript.

**# 7 In all hydrographs, especially in Figure7 and Figure 8, after WRF-Hydro is calibrated for the summer months only, and after the ensemble estimations are made, a substantial mismatch between observations and simulations is seen for streamflow for the months of September until December. What is this underestimation attributed to? Is there a source for**

**streamflow not accounted in the model? Is there a physical process that is missing or is the precipitation forcing inadequate? What explanation can the authors give?**

Response: Thanks for noticing and asking this question. Streamflow is consistently underestimated after the monsoon season (September-December). Several factors contribute to underestimated streamflow during this time of the year. Firstly, the region receives limited precipitation during this time of the year and most of it is in the form of solid precipitation. Therefore, runoff mostly is generated from snow/glacier melt in the headwater. However, WRF has limited capability to fully parameterize the glacier dynamics. In addition, it underestimates the SWE observation, as found in other parts of the world (for example, it underestimates SWE over the Rocky Mountains in the USA), but we do not have sufficient observations in the Himalayas to prove this. Consequently, WRF may not have sufficient snow remaining in the lower elevations beyond summer monsoon due to the underestimation of SWE and/or excess snow melting. The calibration process tries to match the observed streamflow. However, with a slightly underestimated precipitation in case of MP8KF simulations, the snow parameters try to find a value that causes more snow melting to better match the hydrograph peaks during summer, but is left with insufficient snow beyond summertime. We will discuss these aspects in the discussion section of the revised manuscript.

**Other comments**

**L218-225: It is unclear how the LULC update is made.**

Response: Thanks for raising this point. Indeed, we did not provide details about the LULC update to keep the paper concise and focused on the calibration process. However, we understand that readers might be interested to know about this step. Therefore, we will update the LULC processing section in the revised version of the manuscript.

**L297: How is "a change in the hydrograph" defined? Is there a particular measure used?**

Response: Thanks for this question. We just checked the hydrograph by visual inspection and other statistics such as total annual volume and annual mean volume.

**L612-614: Are there previous studies examining the results of ensemble averaging on precipitation and the impact of it on streamflow?**

Response: Thanks for this question. We are not aware of any such studies that use ensembles in the same way that we did. However, we will revisit the literature again and update the discussion section if needed.

**L618-621: Which land use classes used by the land surface component of WRF and WRF-Hydro are there in the Beas basin? What is the fractional coverage? This type of information**

**can be added in the study area description and could be related to the discussion on the sensitivity of parameters.**

Response: Thanks for this suggestion. We have the name of LULC classes that are present in the region, however we do not have fractional coverage information, which would be insightful to understand the region's physiography. We will update the study area and discussion section as per the suggestion in the revised manuscript.

**L651: How would the use of OVROUGHRT and RETDEPRT change the results in the current study?**

Response: Thanks for asking. We are aware that a few studies changed OVROUGHRT and RETDEPRT to adjust the hydrologic response among surface and subsurface fraction. However, we did not change these parameters, as these are just the scaling parameters of surface roughness and surface retention depth. Instead, we exploited the full theoretical range of surface roughness and hydraulic conductivity parameters, allowing them to find their optimum values.

**Equations 1-4: The equations presented in this section fit better in the Methods rather than the results. Could you generalize the equations and move them to Methods?**

Response: Thanks for the suggestion. We will move these equations into the Method section of the revised manuscript.

**Minor comments:**

**L58: The introduction of the Beas basin is abrupt to readers unfamiliar with the study area. What is the importance of the Beas basin relative to the broader region? Please rephrase or move the sentence to a later paragraph when the focus is on the specific basin.**

Response: Thanks for the suggestion. We will update the introduction as per the suggestion by writing the Beas basin introduction in later stage of the paragraph.

**L55-57 and L69-71 are contradictory. If the glaciers disappear by the end of the century by 90%, how will the water stored in glaciers meet the water demands at least by the end of the 21st century? Is there high uncertainty reported in other studies? If yes, this could be mentioned.**

Response: Thanks for asking. In literature there is indeed some contradiction regarding water availability at the end of the 21st century (studies we cited between L50-L70) and it is related to scale. Studies that indicate less vulnerable water availability are focused on larger basins. Conversely, isolated and smaller basins are frequently reported to be vulnerable and are already or

projected to be under water stress. The contrast in the findings based on the basins' size and topographical setting serve as a strong motivation to study smaller and localized basins for their water risk assessment.

In larger basins, some sub-basin that have sufficiently larger areas with elevation higher than snow line equilibrium may retain substantial snowpack to offset warming impact and provide sufficient water supply downstream. Otherwise, smaller and lower elevated regions are not resilient to warming and do not have sustained snowpacks to provide water supply. However, a combination of these 2 sub-regions may have potential to offset the water risk overall.

In addition, L55-57 highlights literature findings in the regions beyond the study area providing contrasting perspectives on water supply within the broader context of the Himalayan region. Meanwhile, L69-71 indicates the glacier change projections within the study region, along with a citation that shows strong dependence of water supply on glacier melt in the study region.

In the revised manuscript, we will make an effort to clarify the underlying reasons for these seemingly contradictory results from prior studies.

**L290: "…having each parameter perturbed to its default value..". Do you mean perturbed from the default value with the range of possible values specified in Table 2?**

Response: Thanks for asking. Yes, it is perturbed *from* its default values and within the range shown in Table 2.

**Table 2: Please provide the units of the listed parameters. If possible, please add the description of these parameters.**

Response: Thanks for the suggestion, we will update this table with these details in the revised manuscript.

**What does (2009) stand for with KGE and what are the units of RMSE in Tables 4-7?**

Response: Thanks for asking. 2009 refer to the year when the paper on this metric was published i.e. Gupta et al. (2009). And, the unit of RMSE is the same as streamflow which is cfs. We will update these details in the revised manuscript.

**L306-309: Please define parameter identifiability?**

Response: Thanks for asking. Parameter identifiability is the variance of a parameter in a truncated solution space. We will add more details on this in the revised version of the manuscript.

**L616: "LSQR has the advantage of computational speed", compared to what?**

Response: Thanks for asking. LSQR has the advantage of computation speed over SVD regularisation in case of highly-parameterised inversion problems. We will rephrase this sentence to be more clear in the revised manuscript.